# Controlling supercurrents and their spatial distribution in ferromagnets

Kaveh Lahabi[1], Morten Amundsen[2], Jabir Ali Ouassou [2], Ewout Beukers[1], Menno Pleijster[1], Jacob Linder[2], Paul Alkemade[3] & Jan Aarts [1]

Spin-triplet Cooper pairs induced in ferromagnets form the centrepiece of the emerging field of superconducting spintronics. Usually the focus is on the spin-polarization of the triplets, potentially enabling low-dissipation magnetization switching. However, the magnetic texture which provides the fundamental mechanism for generating triplets also permits control over the spatial distribution of supercurrent. Here we demonstrate the tailoring of distinct supercurrent pathways in the ferromagnetic barrier of a Josephson junction. We combine micromagnetic simulations with three-dimensional supercurrent calculations to design a disk-shaped structure with a ferromagnetic vortex which induces two transport channels across the junction. By using superconducting quantum interferometry, we show the existence of two channels. Moreover, we show how the supercurrent can be controlled by moving the vortex with a magnetic field. This approach paves the way for supercurrent paths to be dynamically reconfigured in order to switch between different functionalities in the same device.

[1] Huygens—Kamerlingh Onnes Laboratory, Leiden Institute of Physics, University Leiden, P.O. Box 9504 , 2300 RA Leiden, The Netherlands. [2] Department of Physics, Center of Excellence QuSpin, Norwegian University of Science and Technology, NO-7491 Trondheim, Norway. [3] Kavli Institute of Nanoscience, Delft University of Technology, Lorentzweg 1, 2628 CJ Delft, The Netherlands. Correspondence and requests for materials should be addressed to J.A. (email: aarts@physics.leidenuniv.nl)

The conversion of spin-singlet Cooper pairs to the equal-spin triplets which are needed in superconducting spintronics[1,2] requires carefully designed interfaces between a conventional superconductor (S) and a ferromagnet (F). The process entails both spin-mixing and spin-rotation, and can be brought about by magnetic inhomogeneities at the interface[3]. One method to realize this is to place a thin ferromagnet F′ at the S/F interface, and make the magnetization of F and F′ non-collinear[4]. This technique was recently implemented in Josephson junctions described by 1D geometries, where the supercurrent amplitude was controlled by varying degrees of magnetic non-collinearity (MNC)[5–7]. The present letter establishes a different direction. Here, the central goal is to exert dynamic control over the triplet generator and thereby to determine where the supercurrent spatially flows.

We demonstrate how distinct supercurrent paths in a device can be tailored entirely by spin texture, and altered in a dynamic fashion. Such behavior is intrinsically higher-dimensional and can pave the way for novel hybrid devices in superconducting electronics.

## Results

**Micromagnetic simulations**. The device consists of a disk-shaped planar Josephson junction involving a multilayer of Co/Cu/Ni/Nb, as shown in Fig. 1a. A central trench cuts the top superconducting Cu/Ni/Nb layers in two halves, here connected via a Co weak link. The disk design combines two crucial elements. First, the magnetic moments in Co are arranged in plane and orthogonal to the trench between the superconducting electrodes, while the moments in Ni lie also in plane but parallel to the trench. Micromagnetic simulations show that this geometry results in a well-defined magnetic ground state with a high degree of MNC, a condition optimal for generating triplets (Fig. 1c–e). An equally important element is that the disk shape creates a magnetic vortex state in the Co. This vortex produces a distinct suppression of MNC at the centre of the disk (Fig. 1e), which will be used to distribute the supercurrent in Co over two channels. The MNC suppression is due to the local out-of-plane magnetization at the vortex core, which turns the magnetic moments in the Ni also out-of-plane and, hence, collinear to the Co moments. Incidentally, the in-plane exchange field gradient of a magnetic vortex, without a second ferromagnet, has also been proposed to generate long-ranged triplets[8,9].

**Supercurrent calculations**. To investigate whether a supercurrent can be expected, we numerically simulate the critical current density passing through the Josephson junction by solving the quasiclassical Usadel equation[10] in 3D using the magnetization texture obtained from the micromagnetic simulations. We do this by means of the finite element method, using the finite element library libMesh[11] in a similar fashion as in ref. [12] (for details, see Supplementary Note 1, Supplementary Fig. 3). The superconductors are modeled as bulk, with a phase difference of $\Delta\phi = \frac{\pi}{2}$. In Fig. 2a the discretized model is shown. To reduce the calculation time we truncated the otherwise circular geometry to a width of 40% of the disk diameter, as the currents farther away from the trench are negligible. The results are shown in Fig. 2b, c, where it can be seen that the critical current is suppressed at the

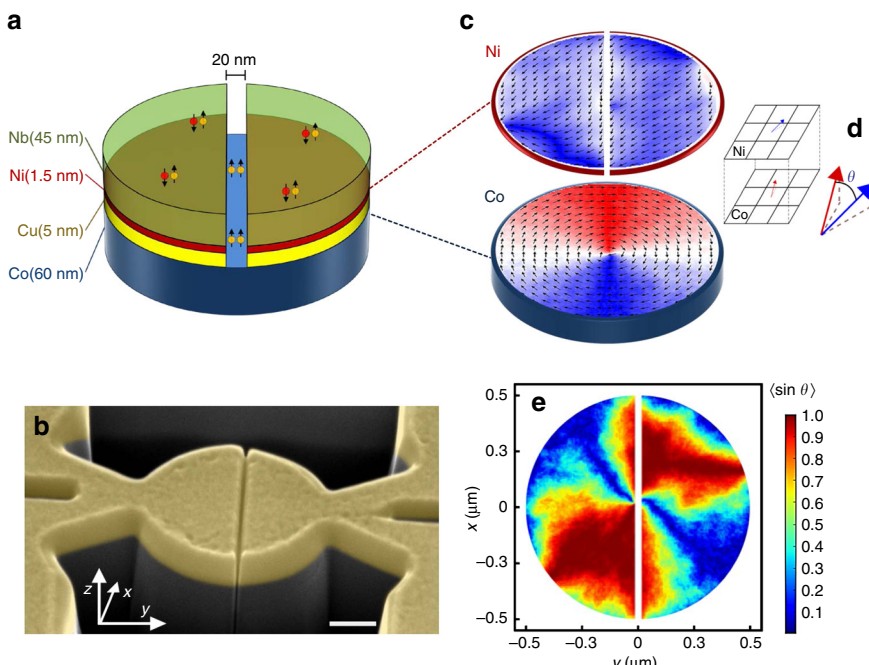

**Fig. 1** Micromagnetic simulations and device layout. **a** Schematic of the device layout. **b** False-color scanning electron microscope image of a device. The scale bar corresponds to 250 nm. The disk is structured with Ga⁺ focused ion beam (FIB) milling. The junction is formed by opening up a gap in the top Nb/Ni/Cu layers, leaving only Co in the weak link (see Methods section for more details). **c** Plane view of the magnetic states of Co and Ni layers in the disk (from 3D OOMMF simulations). The pixel color scheme, red-white-blue, scales with the magnetization along $y$. Magnetic moments in Ni tend to align with the gap which defines the junction, while the vortex configuration in Co arranges the magnetic moments perpendicular to it. This provides a high degree of magnetic non-collinearity (MNC) for triplet generation. The curled magnetic structure of the vortex is also highly effective in minimizing the stray fields from Co, which otherwise would dominate the Ni magnetization, hence compromising our control of MNC. **d** Representation of our method to obtain the MNC profile. For each cell at the top of the Co layer, we determine the angle $\theta$ between its magnetization vector and that of the Ni cell above. **e** Spatially resolved MNC profile calculated from the simulation results shown in **c**. The observed suppression of MNC (the blue region) at the centre of the junction is a result of interlayer dipole coupling at the vortex core

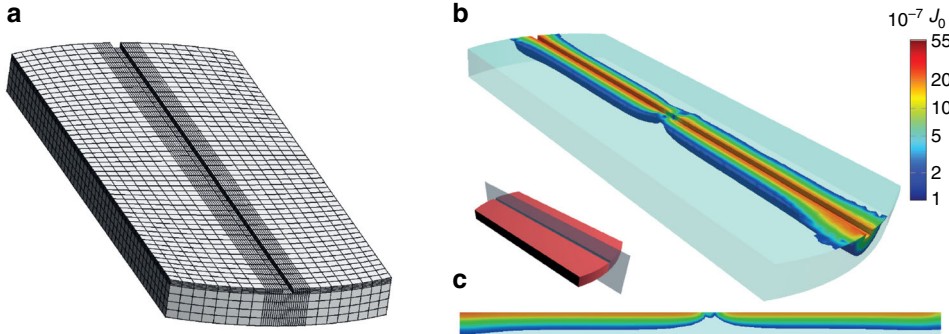

**Fig. 2** Numerical simulation of the critical current. **a** The discretized model (or mesh) used in the numerical simulation of the critical current. Since the triplet current is mostly concentrated in the immediate vicinity of the trench, the mesh density (and hence the accuracy) is set to be higher for this region. For the same reason, the regions farthest away from the trench have been removed to reduce the calculation time. **b** The critical current density divided by a factor $J_0 = \frac{N_0 eD\Delta}{2\xi}$, where $N_0$ is the density of states at the Fermi level, $D$ is the diffusion constant, $\Delta$ is the superconducting gap and $\xi$ is the superconducting coherence length. For clarity, currents lower than $10^{-7}J_0$ are not shown. **c** A slice through the centre of the trench, showing how the current passes across the Co barrier in two separate channels, on either side of the vortex core

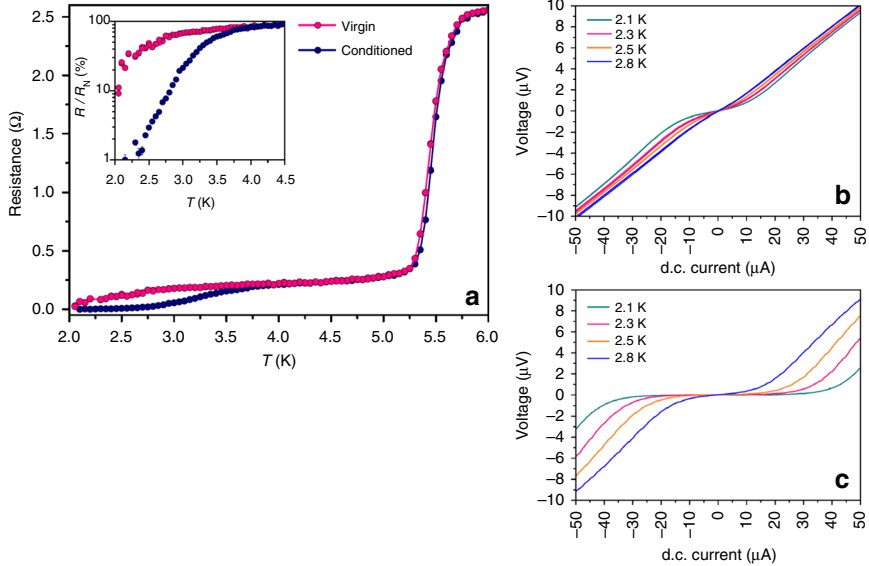

**Fig. 3** Junction transport in the virgin and conditioned states. **a** Resistance as a function of temperature, measured using 10 μA, before (pink) and after (navy) conditioning the sample. Each set shows two distinct transitions. At $T = 5.5$ K, the Nb electrodes become superconducting, while the junction is still in the normal state ($R_N \approx 240$ mΩ). Upon cooling further, resistance undergoes a second transition as the barrier begins to proximize by triplet correlations —eventually reaching zero resistance. For clarity, the $R-T$ dependence at lower temperatures is plotted on a logarithmic scale in the inset. While the superconducting electrodes are unaffected by conditioning the ferromagnets, we observe substantial enhancement of superconductivity in the barrier. **b**, **c** $I$−$V$ traces taken at several temperatures before and after conditioning the sample, respectively. The pronounced contrast between the two sets indicates that transport depends strongly on the magnetic configuration of the junction

centre of the disk, thereby effectively creating two separate current channels.

**Basic transport properties**. As shown in Fig. 3, our junctions show zero resistance and finite critical currents $I_c$ below 3 K. The magnetic state of the sample was conditioned by applying a 2.5 T out-of-plane field at 10 K. This is to reduce the stochastic magnetization introduced by FIB milling when structuring the junction. Figure 3 shows there is a strong difference with data taken before and after conditioning the sample, which is a first indication that MNC and a triplet supercurrent are involved (also see Supplementary Note 2). For instance, conditioning allows the magnetic moments in Ni to rearrange more freely, and align with the gap opened by the FIB. This process increases the MNC in the vicinity of the barrier which, in turn, results in an enhancement of triplet supercurrent at zero field. A consequence of this can be

found in the pronounced contrast between the $I$−$V$ traces measured before and after conditioning the magnetization, as shown in Fig. 3b, c.

**Superconducting quantum interferometry**. To examine the spatial distribution of current density across our junctions, we apply an out-of-plane magnetic field $B_z$, and analyze the resulting supercurrent interference pattern. As demonstrated by Dynes and Fulton[13], the shape of such a superconducting quantum interference (SQI) pattern is given by the Fourier transform of the position-dependent critical current density across a junction $J_c(x)$ through

$$I_c(B_z) = \left| \int_{-R}^{R} dx\, J_c(x)\, e^{2\pi i L B_z x \Phi_0} \right|, \tag{1}$$

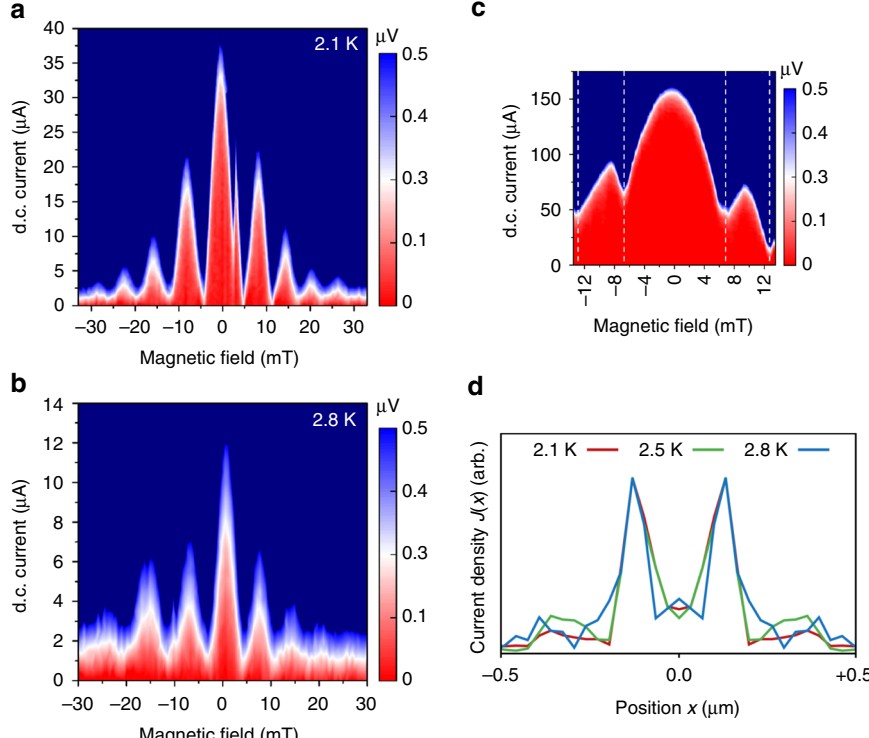

**Fig. 4** Interference patterns and the corresponding current density profiles. **a**, **b** the result of superconducting quantum interference (SQI) measurements taken at 2.1 and 2.8 K, respectively. The patterns show clear double-slit interference, with all lobes having the same width. **c** Single-slit interference pattern from a disk junction where transport is dominated by singlet correlations via a non-magnetic barrier. **d** The current density profiles constructed from the Fourier analysis of SQI patterns taken at 2.1, 2.5, and 2.8 K. The presence of two transport channels, responsible for the SQUID-like interference patterns, is evident

where $L$ is the effective length of the junction, $2R$ is its lateral width (here the disk diameter), and $\Phi_0 = h/2e$ is the superconducting flux quantum. In a typical junction, the uniform distribution of supercurrent density ($J_c(x) = $ constant) leads to the well-known Fraunhofer interference pattern with a sinusoidal current-phase relation given by $I_c(B_z)/I_c(0) \sim |\sin(\pi\Phi/\Phi_0)/(\pi\Phi/\Phi_0)|$. Characteristic for the Fraunhofer pattern is a central lobe that is twice as wide as the side lobes (as in Fig. 4c). These oscillations decay with a $1/B$ dependence. Different device configurations may introduce deviations from the standard pattern, but the described relative widths of the lobes persist as a common feature in all Josephson junctions, since it represents a single-slit interference pattern. In contrast, we expect our disk to exhibit a double-slit interference pattern. This is characterized by slowly decaying sinusoidal oscillations with $\Phi_0$-periodicity, where all lobes have the same width. These patterns are typical for superconducting quantum interference devices (SQUIDs) which, contrary to our device, consist of two individual junctions operated in parallel.

As shown in Fig. 4a, b, the period of the oscillations in our disk device is 7.8 mT (i.e., fluxoid quantization over an effective area of $2.65 \times 10^{-13}$ m$^2$), and appears to be temperature-independent. Qualitatively, the SQI patterns in Fig. 4a, b already foretell the presence of two supercurrent channels: the width of the central lobe is comparable to that of the side lobes, and the oscillations decay far more gradually in field than as $1/B$. Two-channel interference patterns were recently observed in junctions with topological weak links[14–16], where the two-slit interference is a result of edge-dominated transport caused by band bending. In our junction however, this is due to the suppression of triplet supercurrent by the (controllable) magnetic vortex core.

To illustrate the contrast with single-slit interference in a similar device configuration, we prepared a disk junction without the Ni layer, and retaining a thin layer of Cu/Nb at the bottom of the trench. This provides a non-magnetic path in the barrier, allowing singlet correlations to contribute to junction transport. Indeed, we observe a typical Fraunhofer-like interference pattern with a two times wider central lobe, shown in Fig. 4c. Provided that singlet current can dominate the transport, similar results can also be produced in presence of the Ni layer (Supplementary Fig. 5).

Figure 4d shows the supercurrent density profiles extracted from Fourier analysis of the measured interference patterns. A description of this method can be found in the Supplementary Note 3, Supplementary Fig. 4 but it should be mentioned that there is some arbitrariness in choosing the position of the sample edge if the effective junction length $L$ is not known. We put the edge at the position where the current density goes to zero, which leads to a value for $L$ of 170 nm. This is a reasonable number. For a homogeneous junction where $L = 2\lambda_L + d$, with $d$ the gap between the electrodes and $\lambda_L$ the London penetration depth, taking 100 nm for $\lambda_L$ of the Nb, would yield $L$ to be of the order of 200 nm. There is no reason however to expect very close agreement as discussed in Supplementary Note 3. Important is that for any choice of the edge position, two distinct transport channels are clearly visible in the extracted profiles. Comparing these results with the simulations, the supercurrents appear to follow narrower paths, located near the centre of the disk. We attribute this to current crowding effects, in which the neck-shaped contacts and their sharp corners lead to a forward orientation of the currents.

It is important to note that the origin of the two-channel transport in our junction cannot be explained by singlet

supercurrents in a doubly connected path. Direct evidence for this can be found in the SQI measurements taken before conditioning the sample (the virgin state). If two separate current paths had formed unintentionally during fabrication, and allowed singlet correlations to bypass the Co layer via two symmetric channels, then those channels would have already been present before the magnetic state conditioning, and the device would have behaved as a SQUID from the beginning. In contrast, despite several

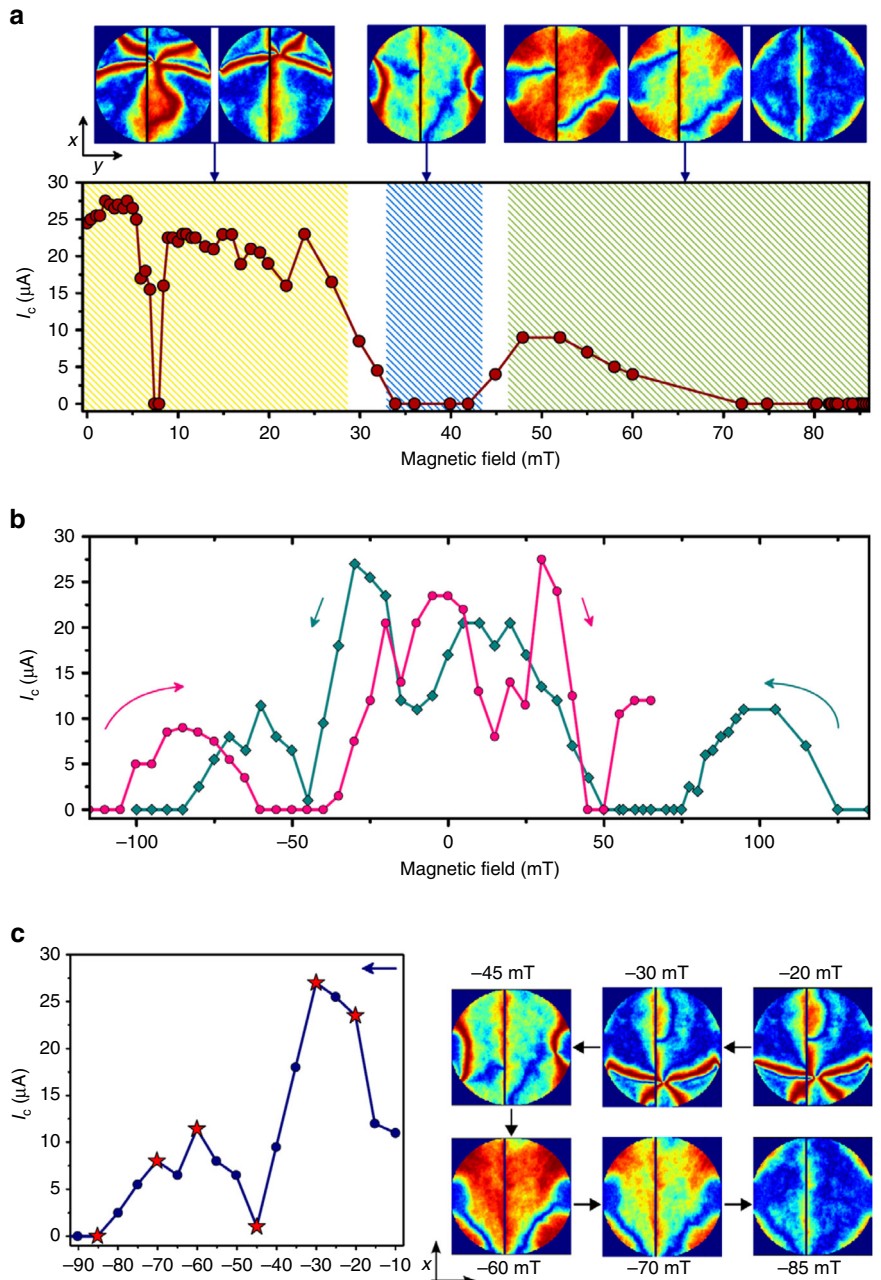

**Fig. 5** Critical current variation and MNC simulations with in-plane field. **a** Measured $I_c$ values and the corresponding magnetic non-collinearity (MNC) profiles, as the system is magnetized by sweeping the field in +y direction. For small fields the vortex core moves along the junction (perpendicular to field direction) to the side of the disk. In this range (shaded yellow), highly non-collinear regions are continuously present and appear to follow the position of the vortex core. The vortex state in Co, which has been effective in suppressing the stray fields, is subsequently removed as the field approaches 30 mT. This leads to a negative dipole field from Co which dominates the effective field acting on Ni. As a result, Ni gets magnetized antiparallel to Co (along −y), hence the suppression of MNC and $I_c$ (shaded blue). As the applied field is raised above 45 mT, it begins to compensate for the local stray fields from the Co layer, ultimately reversing the Ni magnetization along +y. The change in the magnetic orientation associated with this reversal leads to a distinct (re-)emergence of MNC that gradually fades away above 60 mT—as Ni magnetization aligns with Co (shaded green). **b** $I_c$ measured while reversing the field in both directions along y. A clear hysteresis is observed, with individual features are mirrored with respect to field sweep direction. This complex pattern is a result of a changing MNC as the multilayer reverses its magnetization. **c** Positive to negative branch of experimentally measured $I_c(B_y)$ shown together with simulated MNC profiles. Each MNC snapshot is obtained at the specified field, and corresponds to a measurement labeled by the star symbol. Taking steps of 5 mT, simulation shows the vortex enters at −20 mT, moves along −x, and exits the system at −45 mT. The MNC is once again is enhanced at −60 mT, and gradually fades away as the field magnetizes all layers along −y

attempts, no sign of a double-slit interference was found in the virgin state (Supplementary Fig. 1). The SQUID pattern only appeared when the magnetic state was properly conditioned to produce the intended MNC, designed specifically to generate two symmetric triplet channels. More details about the SQI measurements from the virgin state can be found in Supplementary Note 2.

**Magnetotransport measurements with an in-plane field**. Having established the principal role of MNC in shaping the supercurrent, we also examine the possibility of controlling them by altering the MNC profiles using an in-plane field $B_y$ which moves the vortex along the trench. Figure 5a shows the measured currents $I_c(B_y)$ together with the micromagnetic MNC calculations for various stages during the (zero to positive) field sweep. In the first regime (below 28 mT, shaded yellow), we modify the MNC profile by moving the vortex core along $+x$ toward the side of the disk. As the field is raised beyond 30 mT, we remove the vortex, thereby suppressing the supercurrent. The suppression of $I_c$ in this regime (above 34 mT, shaded blue) is caused by the anti-parallel configuration of the ferromagnets, which occurs through the increase of stray fields from Co (now magnetized along $+y$) when the vortex leaves the disk. In the third regime (above 46 mT, shaded green), Ni magnetization begins to reverse from negative to positive $y$ direction, while Co remains magnetized along $+y$. At first, this process recovers $I_c$ as a MNC re-emerges over the entire disk. As we increase the field however, the MNC begins to fade away as both layers magnetize along $+y$, resulting in a gradual suppression of $I_c$. Figure 5b shows the variations in $I_c(B_y)$ when sweeping the field from a high positive to negative value, and back. We observe a complex pattern accompanied with a peculiar hysteresis, where individual features are mirrored (and not just shifted) with respect to the direction of field sweep.

The observed field dependence is fundamentally different from the usual hysteresis in SFS junctions, where the self-field of the ferromagnets can distort or introduce a shift in the interference pattern[7,17,18]. This is rather a distinct characteristic of triplet supercurrents produced by a varying degree of MNC, as the multilayer reverses its magnetization. The measured hysteresis is of a similar nature as the ones reported in refs. [6,7] for multilayer vertical stacks. The most notable difference here is arguably the relatively large field range where $I_c$ is zero, and the pronounced reentrant superconductivity that follows. Figure 5c compares one branch (positive to negative) of the measured $I_c(B_y)$ with the simulated MNC snapshots taken at various stages of the vortex reversal. Even though the experiment and the simulation both sweep the field in steps of 5 mT, the simulated fields for vortex entry and exit translate to direct enhancement and suppression of the measured $I_c$, respectively. For the fields below $-45$ mT, the behavior is similar to the one described for the third regime (green shade) in Fig. 5a.

As a final point, it should be noted that in the present letter we have assumed the channels have an equal phase. This assumption is reasonable for a symmetric MNC (hence spin-mixing) on each side[4]. Whether both channels are 0 or $\pi$, as long as they are symmetric, the SQI results will be indistinguishable. This would not strictly apply to systems with asymmetric spin texture (e.g., caused by vortex displacement), which can result in different phases for the triplet channels[9].

## Discussion
Spin-triplet supercurrents in ferromagnets have been bearing the promise of dissipationless use of spin-polarized currents. This study opens up a completely different direction, in which the focus is not the homogeneous amplitude of the supercurrent, but rather the dynamical control over its spatial distribution. This can lead to novel hybrid devices for superconducting electronics. Moreover, our extensive use of simulations, both of the micromagnetic configurations and of the supercurrents themselves, allow for detailed design and understanding before the actual fabrication of the hybrid device. The next step will be to introduce magnetization dynamics. Magnetic vortices or domain walls can be moved with pulses in the GHz regime, and this can also be simulated. Directing supercurrents then becomes possible on nanosecond timescales, opening the way for high-speed superconducting electronics.

## Methods
**Device fabrication**. Multilayers of Co (60 nm)/Cu (5 nm)/Ni (1.5 nm)/Nb (45 nm) were deposited on unheated SiO$_2$-coated Si substrates by Ar sputtering in an ultra-high vacuum chamber (base pressure below $10^{-8}$ Pa). The thickness of Co and the diameter of the disk (1 μm) are chosen to ensure stabilization of a magnetic vortex[19,20]. The 5 nm Cu layer is used to avoid exchange coupling between the layers. The thickness of the Ni layer was tuned for optimal triplet generation in similar systems[21,22]. The samples were subsequently coated with Pt (7 nm) to protect them from oxidation and to reduce the damage introduced by Ga$^+$ ions during focused ion beam (FIB) processing.

A combination of electron-beam lithography and FIB milling (50 pA Ga$^+$ beam current) was used to structure the disks. Next, FIB with 1 pA current was applied to open the sub-20 nm gap that forms the junction. The trench depth is controlled by the duration of milling. The 1 pA beam current provided sufficient timespan (several seconds) to vary the depth in a controlled manner. The device used for investigating single-slit transport was subject to the same processing steps, with the following exceptions. First, the multilayer was deposited without Ni to minimize triplet generation. Second, when creating the weak link, the duration of FIB milling was reduced by 20% to retain a layer of Cu/Nb at the bottom of the trench. This provides a non-magnetic path for singlet supercurrent in the weak link (on top of Co).

The trench is presumably deeper near the sides of the disks (where sputtered atoms can escape more easily) than at the centre. Hence, in contrast to triplets, singlet correlations would favor the centre of the disk where a non-magnetic channel may be still present on top of the Co.

**Magnetotransport measurements**. The magnetic properties of Co and Ni films used in our devices were characterized by ferromagnetic resonance experiments and SQUID magnetometry. Transport measurements were performed in a Quantum Design Physical Properties Measurement System where samples could be cooled down to 2.1 K. For both in-plane and out-of-plane measurements, the field was aligned within 2° of the sample plane. Resistance versus temperature was measured with a current of 10 μA. The current-voltage characteristics were taken in a four-probe configuration using a current-biased circuit and a nanovoltmeter. The critical current was determined using a voltage criterion: $V > 0.3$ μV for SQI and $V > 0.1$ μV for the measurements with an in-plane field.

The virgin state was measured directly after fabrication (Supplementary Note 2). Prior to the $I_c(B_z)$ measurements presented in the letter, the magnetic state of the sample was conditioned by applying a 2.5 T out-of-plane field at 10 K. The sample was stored in a UHV chamber for 106 days and re-wired to a different puck, and the same measurements were repeated using a different magnet. We were able to reproduce the same $I_c$ patterns, and no discernable changes in transport characteristics (e.g., $R(T)$ or $I_c$) were observed.

**Micromagnetic simulations**. Micromagnetic modeling of the behavior of magnetic Josephson junctions was reported before[23]. Here, finite element micromagnetic calculations were carried out using the Object Oriented Micromagnetic Framework (OOMMF)[24]. The multilayer is divided into a three-dimensional mesh of 5 nm cubic cells. The exchange coefficient and saturation magnetization of Co were set to $30 \times 10^{-12}$ Jm$^{-1}$ and $1.40 \times 10^6$ Am$^{-1}$, respectively, while for Ni these values were $9.0 \times 10^{-12}$ Jm$^{-1}$ and $4.90 \times 10^5$ Am$^{-1}$. The Gilbert damping constant $\alpha$ was set to 0.5 to allow for rapid convergence. The direction of anisotropy was defined by a random vector field to represent the polycrystalline nature of the sputtered films. The Usadel calculations are based on static micromagnetic simulations of a multilayer disk with a diameter of 1 μm. For simulations with an applied in-plane field (shown in Fig. 5), the disk design was extended to include the leads used for transport measurements in the actual device (Supplementary Fig. 2). In the absence of in-plane fields, the overall magnetic configuration remains relatively unaffected by the leads: the vortex core continues to suppress the MNC, resulting in two main channels for long-ranged triplet correlations. However, the influence of the leads on shape anisotropy becomes relevant when sweeping the field along $y$. This allows for an accurate estimate of the MNC, and the resulting variation in $I_c$ during the magnetization reversal.

**Control experiment**. In addition to the device used for investigating the triplet currents, a control sample was prepared in parallel, on the same substrate. This was deposited together with the main device, and received the same treatment, with only one exception: the $Ga^+$ dose used for opening the gap that forms the weak link was lowered by 50%. Reducing the dose stops the milling before it reaches the Co layer in the trench. This leaves a non-magnetic path in the weak link for singlet correlations. The contribution of singlet supercurrent results in a critical current that is around 20 times higher than its neighboring junction (the main device) where the Co weak link can effectively suppress singlet correlations, hence allowing long-ranged triplet supercurrents to dominate the transport. Unlike triplets, the singlet current is not sensitive to the spin texture (i.e., MNC) of the system. This is evident from the single-slit (Fraunhofer-like) interference pattern, shown in Supplementary Fig. 5.

**Data availability**. The data that support the findings of this study are available from the corresponding author upon request.

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

## Acknowledgements

The authors would like to thank S. Goswami, A. Singh, M. Kupriyanov, S. Bakurskiy and J. Jobst for valuable discussions and comments. This work was supported by The Netherlands Organisation for Scientific Research (NWO/OCW), as part of the Frontiers of Nanoscience program. The work was partly supported by the Research Council of Norway through its Centres of Excellence funding scheme, project number 262633, QuSpin. Support was also received from COST actions MP1201 and CA16218.

## Author contributions

K.L. and J.A. conceived the disk geometry, K.L. and E.B. performed the micromagnetic simulations, M.A., J.A.O. and J.L. performed the supercurrent simulations and assisted in the Fourier analysis, K.L., M.P. and P.A. fabricated the devices, K.L. and M.P. performed the measurements. All authors contributed to discussions.

## Additional information

**Competing interests:** The authors declare no competing financial interests.

