## [Peer Review File · Nature Communications]

Reviewers' comments:

Reviewer #1 (Remarks to the Author):

This manuscript reports an experimental study of spin-triplet supercurrents flowing through a Co disk. Due to the magnetic configuration in the disk (circular magnetization with a vortex in the middle), the supercurrent splits into two paths, which can be controlled by moving the vortex with an applied magnetic field. The experiments are carried out well, and theoretical modeling of the magnetic state and its effect on the supercurrent are thorough. In addition, the manuscript is written very clearly. This work represents a new approach to controlling spin-triplet supercurrents, and I believe that the paper contains sufficient novelty to justify publication in Nature Communications.

I have a few minor suggestions for the authors.

- 1) I could find the diameter of the Co disk only in Figure 1(a). It would help to have it also in the text or maybe even in the caption to Figure 1. (It took me quite some time to find it!)
- 2) I am confused by the discussion of the "screening supercurrent" in the first paragraph on page 4. In Josephson junctions with dimensions small compared to λ_J , the magnetic field is not screened from the junction. The non-uniform current distribution in a field is due to the variation of phase across the junction, which follows the vector potential. (See Chapter 4 of the book by Barone and Paterno.) I do not think it is correct to refer to the current distribution derived from the Fraunhofer pattern as a "screening supercurrent."
- 3) On the same topic, the Supplementary Material contains a discussion of the Fourier analysis used to derive the current distributions from the measured Fraunhofer patterns. In that section, the authors state that "the effective junction length is limited by the geometry and not the penetration depth." But further down on that page, they find that the effective junction length is $L = 180$ nm. Since L is considerably less than the disk radius, I would argue that this finding contradicts the earlier statement. (Note that L is very close to the value of $2\lambda_L$ found by other researchers in this field, which is reassuring.)
- 4) The caption to Figure 3 should state what drive current was used in the measurements depicted in Figure 3(a). That will enable a curious reader to check the consistency of the data shown in Figures 3(a) and (b).
- 5) In the Supplementary Material, the numerical calculations of the critical current are described. There it is stated that the thickness of the Ni was set equal to $\xi_s = 10$ nm, the superconducting coherence length. But in the actual samples, the Ni is only 1.5 nm thick. A sentence should be added explaining why it makes little difference to set the Ni thickness to the larger value.
- 6) There is a typo at the very end of the caption for Figure S6: "tanek" -> "taken"

Reviewer #2 (Remarks to the Author):

In this work the authors study a possibility of generation of odd-frequency spin triplet supercurrent in hybrid S/F structures. They use an original approach to fabricate small circular shape planar SF1F2F1S junctions starting from F1F2S trilayer films. The circular shape of the junction and the choice of materials help to achieve the magnetic vortex (MV) state only in the bottom Co layer, while the intermediate Ni layer remains in the in-plane magnetization state. The authors argue that this should lead to modulation of the triplet current within the junction with two maxima of $J_c(x)$. This in turn should change the $I_c(H)$ modulation. Using a combination of micromagnetic simulations and numerical calculations of the critical current, the authors argue that their junctions indeed show such a double maximum. Their second argument is that the critical current is affected by the presence of the MV, which can be removed by applying strong enough in-plane H_y field.

This is an interesting approach to the very interesting problem. Yet, since the magnetic state is not probed directly, I am still missing the hard evidence. The main argument of the authors is the observed half the width of the central $I_c(H_z)$ lobe, explained by the double maxima in $J_c(x)$ profile. The main question is how to prove that it is due to triplet current and not just an artifact of fabrication (which may happen). The answer is rather obvious from presented simulations: if it's the triplet current than the maxima are not fixed and should move together with the vortex. I.e. $I_c(H_z)$ modulation patterns should strongly depend on the presence of an additional $H_{x,y}$ field component. Upon moving of the MV from the center of the structure the two maxima should be strongly altered both in position and amplitude. When the MV is not in the middle (at finite H_y), the maxima should become dissimilar. This should strongly affect the modulation pattern. One way to study this is to measure $I_c(H)$ in inclined field, which would offset the MV from the center and thus should strongly affect the shape of the $I_c(H_z)$ pattern. Unfortunately such a data is not presented. I suppose that if the authors add such data and it would be in agreement with their arguments than the work would be suitable for publication.

Additional comments:

What can be said about sample alignment? Could oscillation of $I_c(H_y)$ be just due to flux quantization in the out-of-plane field component?

In Fig.4 top panels (simulations), do horizontal lines correspond to the junction? It is rotated by 90deg with respect to other sketches. Is the field along x or y? Vortex should move in the direction of the field isn't it? Did you try to apply field in the x direction. $I_c(H_y)$ and $I_c(H_x)$ patterns should be significantly different. This could provide an additional argument.

I do not understand the explanation why the maxima rapidly vanish towards the edges: "As the size of the disk-shaped electrodes decreases towards the edges of the junction ($x \rightarrow \pm R$), screening becomes less effective, and the supercurrent density diminishes." I may provide a counter argument: the effective magnetic thickness decreases at the edges and the effective field (flux) reduces. However, even this is not important: Supercurrent density is just a material property. It does not depend on screening. The period of oscillation does, but not the J_c . To my opinion the explanation doesn't work.

Response to the referees

We have carefully considered the comments of the referees concerning our manuscript ‘Controlling supercurrents and their spatial distribution in ferromagnets’. As a consequence, we decided to perform another set of experiments, in order to allow us to underpin our message more clearly, and to be better able to respond to the questions of one of the referees. In the end, the new data are not part of the manuscript, for reasons we will explain. Still, we have made a number of revisions in the description and presentation of the data, which will be detailed below.

Before doing so, we would like to remark that we felt the questions and comments of the referees constructive and fair. The first referee actually signalled ‘sufficient novelty’ and only had what he or she called minor suggestions. The second referee would have liked to see more data, in particular in inclined fields. This was a very interesting suggestion, but we realized rather early on, when discussing the protocol for such measurements, that there is a catch which the referee had probably not foreseen. His or her other suggestion was to measure with an in-plane field parallel to the junction $B \parallel x$ (perpendicular to the trench). This was successfully carried out by remounting the sample in a (90°-)rotated configuration. We found no changes in the transport properties of the device (e.g. I_c or T_c), and the same double-slit SQI pattern could be reproduced with an out-of-plane field. The sample was then rotated in-plane to measure $I_c(B_x)$. As expected, the I_c patterns are entirely different from the ones measured for $B \parallel y$. This result not only confirms our original explanation, but also indeed provides an additional argument for the way we describe the working of the device, as the referee noted. The physics, however, appears to lead in a very different direction which is rather for a follow-up report. We will gladly share some of those data with both referees (they will be found below), but we chose not to incorporate them in the manuscript at this point.

One more general remark we want to make with respect to the comments of the referees is that both of them had questions with respect to our discussion of the current density profiles obtained from the Fourier transform. We agree that the discussion contained some possibly confusing statements, so this part of the manuscript was rewritten, and we also decided to put data from a control sample in the Supplementary Information to provide additional insight.

Below we will respond to the comments point by point. In the revised manuscript, new text is written in blue.

Referee 1

1) I could find the diameter of the Co disk only in Figure 1(a). It would help to have it also in the text or maybe even in the caption to Figure 1. (It took me quite some time to find it!)

The diameter has been put in text and caption

2) I am confused by the discussion of the “screening supercurrent” in the first paragraph on page 4. In Josephson junctions with dimensions small compared to λ_J , the magnetic field is not screened from the junction. The non-uniform current distribution in a field is due

to the variation of phase across the junction, which follows the vector potential. (See Chapter 4 of the book by Barone and Paterno.) I do not think it is correct to refer to the current distribution derived from the Fraunhofer pattern as a “screening supercurrent.”

The referee is correct. The phrase ‘screening supercurrent’ is only appropriate for long junctions where it basically describes the Meissner effect. For short junctions there are circulating supercurrents, but they do not ‘Meissner-screen’. We inadvertently mixed ‘screening’ and ‘circulating’. There were, however, several reasons to rewrite this discussion more extensively, as will be addressed in the next point.

3) On the same topic, the Supplementary Material contains a discussion of the Fourier analysis used to derive the current distributions from the measured Fraunhofer patterns. In that section, the authors state that “the effective junction length is limited by the geometry and not the penetration depth.” But further down on that page, they find that the effective junction length is $L = 180$ nm. Since L is considerably less than the disk radius, I would argue that this finding contradicts the earlier statement. (Note that L is very close to the value of $2\lambda_L$ found by other researchers in this field, which is reassuring.)

This comment on the effective junction length concerns an issue similar to the one raised by Referee 2 with respect to the supercurrent density distribution as it follows from the Fourier analysis and requires a longer discussion. We agree that the statement was confusing, and we have substantially rewritten that part of the manuscript, including the Supplementary material. The gist of the change is as follows.

An important aspect of the Dynes-Fulton approach is that there is no formal bound on the position along the junction (the x -axis in eq.1). For a rectangular and essentially homogeneous junction, this is solved by reading off the value of L from the first minimum in $I_c(B)$. The current distribution then trivially goes to zero at the sample edge. In our case there is clearly no reason to assume a homogeneous junction. Instead, we choose the position of the edge, and then find an *effective* value of L , since the conjugate to the position x in Fourier transform of $I_c(B)$ is the factor $2\pi LB / \Phi_0$. If we were to take the sharp drop in the current density profile as the sample edge, L would become less than 100 nm, which appears to be too low in view of the value of λ . In either case, however, the distribution is peaked, with a strong minimum at the sample center, which is the main message of the analysis.

4) The caption to Figure 3 should state what drive current was used in the measurements depicted in Figure 3(a). That will enable a curious reader to check the consistency of the data shown in Figures 3(a) and (b).

The drive current (10 μ A) has been inserted in the Measurements section and in the caption of Fig.3.

5) In the Supplementary Material, the numerical calculations of the critical current are described. There it is stated that the thickness of the Ni was set equal to $x_{i_s} = 10$ nm, the superconducting coherence length. But in the actual samples, the Ni is only 1.5 nm thick. A sentence should be added explaining why it makes little difference to set the Ni thickness to the larger value.

This has been done. The main reason to use ξ is to avoid smaller elements, which would much increase the calculation time. This yields lower values for the triplet current, but still serves the purpose of our calculation, namely to identify the origin of the supercurrents.

6) There is a typo at the very end of the caption for Figure S6: “tanek” -> “taken”

This has been corrected.

Referee 2

Comment : I am still missing the hard evidence. The main argument of the authors is the observed half width of the central $I_c(Hz)$ lobe, explained by the double maxima in $J_c(x)$ profile. The main question is how to prove that it is due to triplet current and not just an artifact of fabrication (which may happen).

These are valid remarks, all centering on ‘proving the triplet current’, which deserve several answers.

(i) Regardless of the shape of $J_c(x)$ and its physical origin, the most distinctive signature of triplet supercurrent is its sensitivity to magnetic non-collinearity (MNC). The pronounced contrast in the amplitude of critical current before and after conditioning the sample (shown in Supplementary Fig. S1) is a strong indication for triplets. This remark was hidden in the Methods section, we now make it when describing the $I_c(B)$ data.

Furthermore, if the two channels were to somehow form by accident during fabrication, they should have also been present before conditioning the sample. Despite repeated attempts, we observed no SQUID-like interference pattern when the system was in its virgin state (Fig. S2). The double-slit interference only appeared (and continued afterwards) when the magnetic state was properly conditioned to produce the desired MNC. This remark has been added to the Supplementary.

(ii) Other evidence against an artefact of fabrication comes from a control sample we prepared at the same time and on the same substrate as the sample already presented. Figure 1 shows out-of-plane measurements on the control device. The junction received the same treatment with one exception: the Ga^+ dose used for opening the gap that forms the junction was reduced by 50%. This leaves a nonmagnetic layer in the weak link. Singlet correlations therefore dominate the junction transport, resulting in a critical current that is approximately 20 times larger. The interference pattern is Fraunhofer-like, with a two times wider central lobe. We incorporated the results on this control sample in the Supplementary information, while briefly mentioning its fabrication in the Methods section of the main text.

Comment : (how to prove triplet currents). The answer is rather obvious from presented simulations: if it is the triplet current then the maxima are not fixed and should move together with the vortex. I.e. $I_c(H_z)$ modulation patterns should strongly depend on the presence of an additional $H_{x,y}$ field component. Upon moving of the MV from the center of the structure the two maxima should be strongly altered both in position and amplitude. When the MV is not in the middle (at finite H_y), the maxima should become dissimilar. This should strongly affect the modulation pattern. One way to study this is to measure $I_c(H)$ in inclined field, which would offset the MV from the center and thus should strongly affect the shape of the $I_c(H_z)$ pattern. Unfortunately such data is not presented.

We agree that displacing the vortex would alter the supercurrent distribution. Observing that with the Fourier method of Dynes and Fulton is not that simple, however, since the method is based on the assumption that the shape of the distribution is symmetric, $J_c(-x) = J_c(x)$. If $J_c(x)$ contains an asymmetric component, the Fourier method can no longer provide a unique and reliable solution. For instance, an asymmetric $J_c(x)$ would result in an increase of the minima in $I_c(B_z)$ (i.e. lifting the nodes in the interference pattern). The same effect however, can also arise from a non-sinusoidal current-phase relation (CPR), making the analysis less certain by imposing further assumptions about the CPR. (for more details see Supplementary information in Ref [Allen et al, Nature Physics 12, 128–133 (2016)]).

There is another rather fundamental reason that severely limits the information an inclined field can provide. The idea is to use the in-plane (IP) component of the inclined field to change the position of the core, while obtaining an interference pattern from the out of plane (OP) component as we sweep field. The predicament arises from the fact that sweeping the field at a constant inclination angle would also move the vortex by varying the IP component, since the fields to probe and the fields to move (about 20 nm / mT) have similar values. As an example, consider a 30° tilt. The out-of-plane (OP) field required to observe of the order of 4 lobes (needed for the Fourier analysis) is about 25 mT. This requires a total field of 29 mT. However, by the time the field has reaches 29 mT, the IP component has changed from 0 to 14 mT. During the OP-measurement, the vortex is therefore continuously shifting its position over a large fraction of the disk (over 200 nm). Even at 10°, the OP component reaches 25 mT when the IP component is 4.4 mT. This corresponds to a vortex displacement of less than 100 nm, which occurs continuously during the sweep. This may be too small to observe in the analysis, but more importantly, the shifting position will blur the outcome of the Fourier transform. Tilt angles above 60° lead to another problem: the IP field pushes the vortex out before the OP component can reach 25 mT (the vortex is removed by IP fields of 35 mT or higher).

We therefore did not perform this experiment. We did however remount the sample in a 90° rotated configuration to measure $I_c(B_x)$. This is discussed below.

Additional comments: What can be said about sample alignment? Could oscillation of $I_c(H_y)$ be just due to flux quantization in the out-of-plane field component?

Alignment: <1° for OP and ±1° for IP measurements. The smallest features in the IP measurements occur over a field range of at least 20 mT. In order for the fluxoid quantization of the out plane component to play a noticeable role, the sample would have to be at least 30° misaligned. Also, in a misaligned field the $I_c(B_y)$ pattern would follow the

shape of the out of plane interference pattern. In contrast, the pattern shows an entirely different symmetry where individual features are *mirrored* with respect to the direction of field sweep. The observed hysteresis is of similar nature to the one reported in Ref. [Banerjee et al, Nat Comm, 4771 (2014)] and is a result of a varying degree of MNC during magnetization reversal. This is fundamentally different from the usual hysteresis in SFS junctions, where the self-field of the ferromagnets can introduce a shift in the interference pattern. We now provide the values for the (mis)alignment in the manuscript.

In Fig.4 top panels (simulations), do horizontal lines correspond to the junction? It is rotated by 90deg with respect to other sketches. Is the field along x or y?

Yes, that is correct. They are / were rotated by 90^0 with respect to the other sketches but we changed that to avoid confusion. The field is along y, as stated in the caption. We inserted an additional x,y-axis sketch in Fig.4.

The vortex should move in the direction of the field isn't it? Did you try to apply field in the x direction. $I_c(H_y)$ and $I_c(H_x)$ patterns should be significantly different. This could provide an additional argument.

No, the vortex moves perpendicular to the applied field. This has been demonstrated in literature (see e.g. Wren et al, JAP **118**, 023906 (2015) and is also the case in our simulations. We made it explicit in the text that B_y moves the vortex along the trench.

Otherwise, this was a logical remark, and we followed the referee's suggestion. The results are presented below in Figs 2. As anticipated by the referee, $I_c(B_x)$ behaves very differently from $I_c(B_y)$. After conditioning the sample, we begin with maximum I_c at zero field and increase the field to 200 mT. The supercurrent is rapidly suppressed as the vortex begins to move away from the junction, now perpendicular to the trench. Remarkably, I_c remains heavily suppressed even when the field is brought back to zero. The junction regains its maximum I_c when the sample is conditioned again.

[REDACTED]

We emphasize that the observed behaviour cannot be accounted for by singlet correlations in accidentally formed current paths. Had that been the case, the current paths would have been constantly present and available for transport, regardless of how the system is magnetically conditioned. The fact that I_c is almost entirely determined by the magnetic history of the system is a clear indication of triplet supercurrent, generated by the MNC.

The origin of the observed suppression of I_c , when reversing from positive to negative fields, however turns out to be more subtle than simply the lack of MNC. Simulations with an in-plane field indicate that, during magnetization reversal, the vortex is not simply brought back, but that an intermediate configuration exists with up to three vortices which converge into a single vortex state at negative magnetic field. This corresponds to the -45 mT in Fig.2, where the critical current recovers its maximum value. The SQI measurements (with an out-of-plane field) from this 'suppressed current' state, indicate that now the *phase differences* between different channels are starting to play a role. Within this intermediate state, $I_c(B_z)$ behaves similarly to the ones measured in the virgin state: it has two maxima and is suppressed around zero field. Remarkably, this trend can be fully reversed by re-conditioning the magnetization. As we switch back to the original state (with a single vortex) we bring back the SQUID pattern with maximum I_c appearing at zero field.

In the first version of the manuscript we did not address the fact that when the Ni magnetizations are orthogonal to the Co, but parallel to each other, they actually form a π -junction and the two channels are like a π, π SQUID. Other configurations will lead to the equivalent of a $0, \pi$ SQUID, where the current is suppressed at 0 field. This is the different physics direction we alluded to above, which goes well beyond the scope of present manuscript. It is important, however, to at least touch upon this issue, so we added the following sentence:

As a final point, it should be noted that in the present letter, we have assumed the channels have an equal phase. This assumption is reasonable for a symmetric MNC (hence spin-mixing) on each side [4]. Whether both channels are 0 or π , as long as they are symmetric, the SQI results will be indistinguishable. This would not strictly apply to systems with asymmetric spin texture (e.g. caused by vortex displacement), which can result in different phases for the triplet channels.

The bottom line is that we can extract widely different critical current behaviour of the device, determined entirely by its magnetic (pre-)history. While the present manuscript cannot fully touch upon all available modes of operation, corresponding to different magnetic states, we hope this can convince the referee we are dealing with triplet supercurrents, generated by the controlled magnetic configuration of the system.

I do not understand the explanation why the maxima rapidly vanish towards the edges: "As the size of the disk-shaped electrodes decreases towards the edges of the junction ($x \rightarrow \pm R$), screening becomes less effective, and the supercurrent density diminishes." I may provide a counter argument: the effective magnetic thickness decreases at the edges and the effective field (flux) reduces. However, even this is not important: Supercurrent density is just a material property. It does not depend on screening. The period of oscillation does, but not the J_c . To my opinion the explanation doesn't work.

Also spurred on by Referee 1, we rewrote this paragraph since the word screening and the ensuing interpretation was not correct, as we discussed above.

REVIEWERS' COMMENTS:

Reviewer #1 (Remarks to the Author):

The authors have adequately addressed the concerns of both reviewers. I find the evidence for spin-triplet supercurrent in their device to be strong and convincing. Given the novelty of this approach, I recommend publication in Nature Communications. I have two minor suggestions:

- 1) On page 3, just below Eqn. (1), one reads, "R is its lateral width (here the disk radius)..." That should be changed to "2R is its lateral width (here the disk diameter)..."
- 2) The new Figure 3(e) should have the temperature labelled on the figure as is done in Figures 3(c) and (d), and also in the figure caption.

Reviewer #2 (Remarks to the Author):

The authors have made proper changes and answered my questions in a satisfactory manner. Although there are still questions left, however, the manuscript does present an important development in the field. Therefore, I recommend publication of the manuscript in its present form in Nature Communication.

Response to the referees

We are very pleased to find that the referees find their earlier concerns adequately addressed, and that they both advise publication. We have made the two (typographic) corrections pointed out by Reviewer 1.

REVIEWERS' COMMENTS:

Reviewer #1 (Remarks to the Author):

The authors have adequately addressed the concerns of both reviewers. I find the evidence for spin-triplet supercurrent in their device to be strong and convincing. Given the novelty of this approach, I recommend publication in Nature Communications. I have two minor suggestions:

- 1) On page 3, just below Eqn. (1), one reads, "R is its lateral width (here the disk radius)..." That should be changed to "2R is its lateral width (here the disk diameter)..."
done
- 2) The new Figure 3(e) should have the temperature labelled on the figure as is done in Figures 3(c) and (d), and also in the figure caption.
done

Reviewer #2 (Remarks to the Author):

The authors have made proper changes and answered my questions in a satisfactory manner. Although there are still questions left, however, the manuscript does present an important development in the field. Therefore, I recommend publication of the manuscript in its present form in Nature Communication.